# Functional and structural segregation of overlapping helices in HIV-1

**Maliheh Safari[1], Bhargavi Jayaraman[1], Henni Zommer[1], Shumin Yang[1,2], Cynthia Smith[1], Jason D Fernandes[1]\*, Alan D Frankel[1]\***

[1]Department of Biochemistry and Biophysics, University of California, San Francisco, San Francisco, United States; [2]School of Medicine, Tsinghua University, Beijing, China

**Abstract** Overlapping coding regions balance selective forces between multiple genes. One possible division of nucleotide sequence is that the predominant selective force on a particular nucleotide can be attributed to just one gene. While this arrangement has been observed in regions in which one gene is structured and the other is disordered, we sought to explore how overlapping genes balance constraints when both protein products are structured over the same sequence. We use a combination of sequence analysis, functional assays, and selection experiments to examine an overlapped region in HIV-1 that encodes helical regions in both Env and Rev. We find that functional segregation occurs even in this overlap, with each protein spacing its functional residues in a manner that allows a mutable non-binding face of one helix to encode important functional residues on a charged face in the other helix. Additionally, our experiments reveal novel and critical functional residues in Env and have implications for the therapeutic targeting of HIV-1.

## Editor's evaluation

Fernandes et al., ask the question: "What are the evolutionary constraints on genomic sequence that encode two different proteins?" To this end, they compare the functional constraints on mutations in HIV Rev and Env, which are encoded in different reading frames from the same region of the viral genome. Interestingly, residues that are functionally constrained in one protein are, for the most part, not as constrained in the other. The elegance of this solution is attractive and will be of interest to the protein evolution and structure communities.

## Introduction

HIV-1 contains nine genes encoded within a relatively short genome of 10,000 nucleotides (*Frankel and Young, 1998*). Eight of these nine genes overlap with another on the DNA level and consequently encode protein products in different reading frames. HIV-1 Env overlaps with Vpu, Rev, and Tat, and some parts of Env overlap with both Rev and Tat. Despite the prevalence of overlapping genes, the evolution and function of overlapping genes remains understudied. However, evidence establishing the importance of overlaps continues to grow: the SARS-CoV-2 pandemic has demonstrated the importance of identifying and studying overlapping reading frames, as many immune epitopes targeting canonical and non-canonical overlapping ORFs (open reading frames) have been identified in patients (*Nelson et al., 2020*; *Weingarten-Gabbay et al., 2020*;; *Finkel et al., 2021*).

We previously showed that, in the case of the essential regulatory genes *tat* and *rev*, HIV-1 dedicates the nucleotide sequence to one protein or the other so that critical functional residues do not overlap with one another (*Fernandes et al., 2016*). For *tat* and *rev*, this arrangement is possible because both proteins consist of functional motifs spaced by disordered linkers, and thus functional motifs of one protein can be encoded in regions that overlap with disordered linkers in the other.

**\*For correspondence:**
jason@scribetx.com (JDF);
frankel@cgl.ucsf.edu (ADF)

Although the vast majority of overlapped proteins are predicted to be disordered (*Rancurel et al., 2009*) and thus have the potential for this arrangement, it is possible that some overlapped regions have more stringent structure and function requirements. Indeed, HIV-1 contains an intriguing overlapping region, between the Rev and Env genes (*Figure 1A*), in which both protein segments are structured and functionally important. This region might represent a viral Achilles' heel, vulnerable to therapeutic targeting, as the mutational landscape for escape mutations would be constrained by the simultaneous requirements of both proteins.

Rev is required for the nuclear export of partially spliced and un-spliced viral RNAs that encode necessary late stage viral proteins as well as the genomic RNA for encapsidation (*Malim and Cullen, 1991*; *Pollard and Malim, 1998*; *Rausch and Le Grice, 2015*). Notably, Rev contains a quasi-helical nuclear export sequence (NES), that binds to the NES-binding groove of the host nuclear export factor Crm1 (*Fornerod et al., 1997*; *Booth et al., 2014*). Env is expressed as a polyprotein, composed of the gp120 surface and gp41 transmembrane proteins, which forms a trimeric viral envelope that mediates viral particle binding to cell entry receptors (*Checkley et al., 2011*; *Murphy et al., 2017*). The cytoplasmic tail of gp41 is the most C-terminal portion of Env and is highly variable in size (*Tedbury and Freed, 2015*). In contrast to other retroviruses, lentiviruses such as HIV-1 have an extended cytoplasmic tail containing the Kennedy epitope (KE) followed by three membrane associated lentiviral lytic peptide motifs (LLP-1, LLP-2, and LLP-3) (*Steckbeck et al., 2013*; *Figure 1A*). HIV-1 LLP-1, and LLP-2 contain highly conserved arginine and lysine residues that are involved in cation-π interactions and have been suggested to stabilize the cytoplasmic tail α-helical structure (*Murphy et al., 2017*). LLPs are amphipathic α helices that embed in the virus membranes (*Kalia et al., 2003*; *Steckbeck et al., 2011*; *Murphy et al., 2017*). LLP-2 may be transiently exposed on the virus surface to interact with the gp41 core and regulate virus-cell membrane fusion (*Lu et al., 2008*; *Murphy et al., 2017*; *Piai et al., 2020*). A recent nuclear magnetic resonance spectroscopy study revealed that LLP-2 forms two amphipathic helices that wrap around the transmembrane domain of Env, arranging a 'baseplate' that supports the transmembrane domain and the rest of the Env trimer (*Piai et al., 2020*). Additional evidence suggests that the C-terminal tail makes critical contacts with the matrix protein (*Alfadhli et al., 2019*) to facilitate Env incorporation into virions.

The Rev NES and a portion of the Env LLP-2 are encoded by the same nucleotides in the viral genome, with Env encoded in the +1 reading frame relative to Rev (*Figure 1A*). As both these domains have important functions in viral replication and are structured, they provide an ideal system to examine how viruses balance the evolution of these two genes with critical, overlapping structured motifs. Here, we perform conservation analyses, functional assays, and selection experiments to examine how the virus balances selection pressures within the Rev-Env overlap. Surprisingly, we find that functional residues are also segregated at the nucleotide level, as for the Rev-Tat overlap. In Rev-Env case, the two-dimensional segregation of nucleotides in the genome is mirrored in the three-dimensional functional surfaces of the proteins such that the nucleotides encoding a functional surface in the helix of one protein encode residues on the non-binding face of the overlapping helix. Furthermore, our analyses reveal three important charged residues on the charged surface of LLP-2 that are required for Env function.

## Results

### Conservation of chemical properties in the Rev NES and Env LLP-2

The Rev NES (residues 75–83) binds to Crm1 using several critical hydrophobic residues and non-canonical prolines that change the helical register and position hydrophobic side chains on one face of the helix to fit within the NES recognition groove (*Dong et al., 2009*; *Güttler et al., 2010*; *Figure 1B*). The Env LLP-2 forms an amphipathic helix of about 33 amino acids long (*Eisenberg and Wesson, 1990*; *Kalia et al., 2003*; *Murphy et al., 2017*) that spans residues 767–788 of Env (242–274 of gp41) (*Figure 1A*). Nine of the LLP-2 residues (767–776) overlap with the quasi-helical Rev NES (*Figure 1B*). Mutating charged residues within LLP-2 or hydrophobic residues in the Rev NES negatively impacts viral replication (*Kalia et al., 2003*; *Newman et al., 2007*; *Güttler et al., 2010*; *Fernandes et al., 2016*), prompting us to examine in detail the selective forces that guide the evolution of this overlapping region.

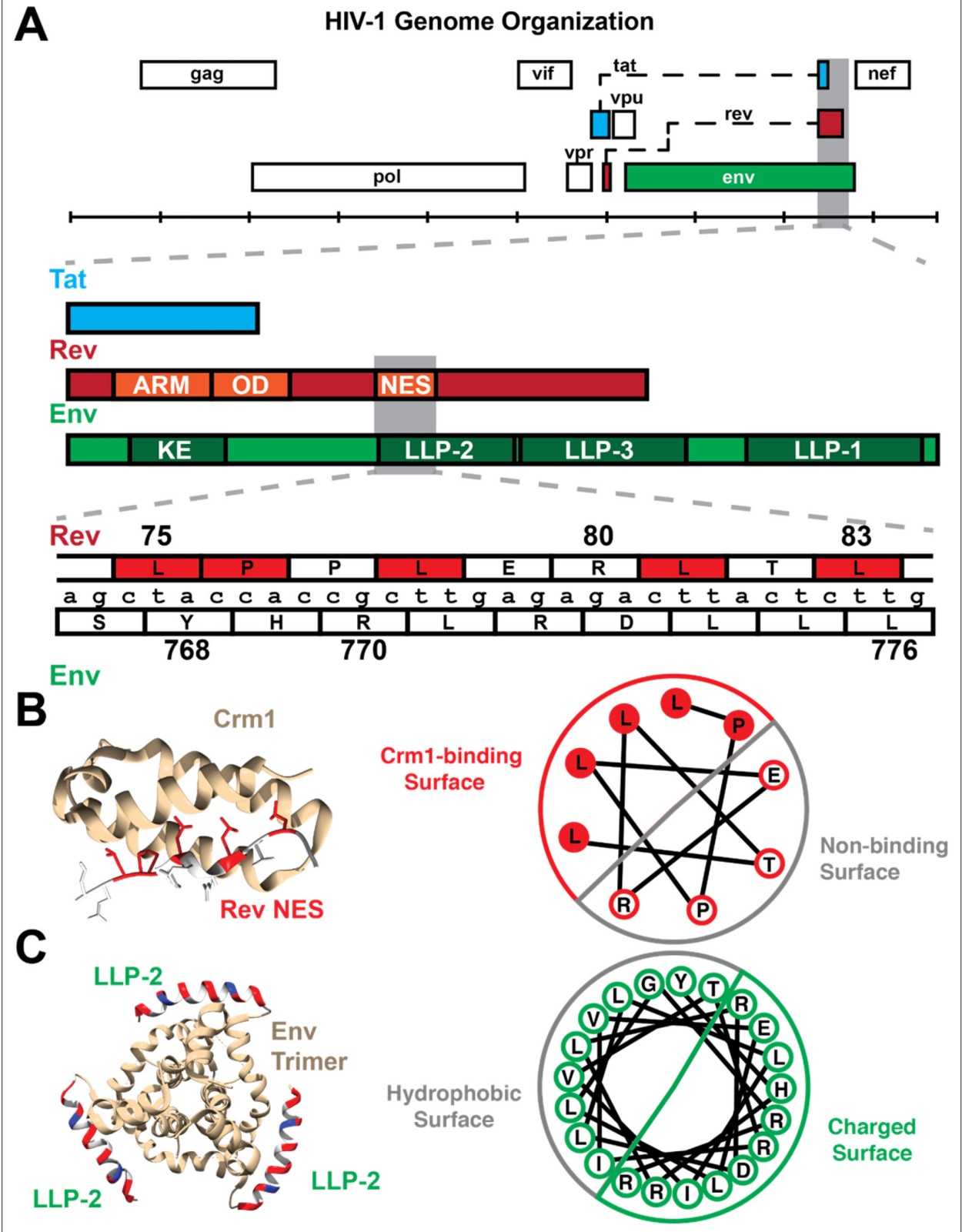

**Figure 1.** Organization of Rev NES/Env LLP-2 overlap. (**A**) Genetic organization of the *rev/env* overlap. (Top) The full genome organization of HIV-1 (HXB2 numbering) demonstrates that the second exon of *rev* overlaps *tat* (second exon) and *env*. A detailed annotation of this overlap depicts the overlap between the quasi-helical NES of Rev with the helical LLP-2 of Env. The ARM (arginine-rich motif, nuclear localization sequence), OD (oligomerization domain), and NES (nuclear export sequence) are other functional domains of Rev that overlap with Env-gp41 (cytoplasmic tail),

*Figure 1 continued on next page*

*Figure 1 continued*

including the KE (Kennedy epitope) and two lentiviral lytic peptide motifs (LLP-3 in addition to LLP-2). (**B**) The structure of the Rev NES (red: binding residues, white: non-binding residues) bound to Crm1 (yellow) (PDB: 3nc0). A helical wheel diagram demonstrates how the arrangement of the 2D amino acid sequence creates a distinct binding surface in 3D. (**C**) Composite modeled structure (PDB: 6ujv) of the Env C-terminus trimer, including the first 21 residues of the LLP-2. The LLP-2 is labeled with the hydrophobic face in gray and the charged surface (positively charged residues in red and negative in blue). A corresponding helical wheel diagram derived from *Murphy et al., 2017*, for the Env LLP-2 is also shown with charged residues colored. Functionally important residues of Rev are labeled in red.

In order to measure the relative conservation of these regions within HIV patient sequences, we first downloaded curated alignments of the Los Alamos Database (hiv.lanl.org) (*Foley, 2013*) and calculated Shannon entropies for each amino acid in both proteins within the NES/LLP-2 overlap (*Pan and Deem, 2011*; *Fernandes et al., 2016*). Sites with low entropy (0=perfect conservation) indicate strong conservation of the amino acid in that particular protein, but it is impossible to determine if the observed conservation is simply driven by codon constraints imposed by the alternative frame. To compare the amino acid conservation of corresponding sites in both proteins, we grouped amino acids that share two nucleotides in the genome and plotted their entropies against one another (*Fernandes et al., 2016*; *Figure 2A*). Notably, the one site that contributes to both functional surfaces (H769 in Env and P76 in Rev) is relatively well conserved in both proteins whereas the one site that contributes to neither interface (L755 in Env and T82 in Rev) is weakly conserved (*Figure 2A*). Unexpectedly, two known essential Crm1-binding residues (L78 and L83) displayed low conservation (high entropy) in the Rev reading frame, but the consensus sequence (*Figure 2B*) indicates that chemically conservative isoleucine substitutions are frequently found, thus explaining the high entropy. Moreover, entropy analysis using a reduced amino acid alphabet in which the chemically similar amino acids I/L/V are grouped produced low entropy values for all Crm1-binding residues (*Figure 2C*). The use of chemically conservative side chains to form critical contacts drastically increases the number of 'synonymous' codons (e.g. the six leucine codons can substitute with the three isoleucine codons with essentially equal fitness) and may reflect the kinds of interaction motifs that are most suited to evolve in constrained overlapping regions. The three charged residues (*Figure 2C*; green dots) in LLP-2 were relatively strongly conserved (*Figure 2B*) but showed lower conservation (high entropy) than the Rev NES-binding residues, suggesting that Rev may more greatly influence the selection of amino acids in the overlapping region.

## Deep mutational scanning of non-overlapped viruses selects for important LLP-2 and NES residues

In order to deconvolute the functional constraints for every residue in both proteins absent the overlap, we created libraries of viruses in which Rev and Env were decoupled from one another. Although HIV-1 *env* has been exposed to deep mutational scanning previously, the overlap with *rev* has prevented exploration of the LLP-2 (*Haddox et al., 2016*; *Haddox et al., 2018*). Briefly, we used the NL4-3 *rev-in-nef* virus in which *rev* is moved to the non-essential, single-frame *nef* locus (*Fernandes et al., 2016*), and the endogenous *rev* locus is ablated, thus allowing the endogenous Env LLP-2 region and the rev-in-nef NES region to evolve independently from one another. We individually randomized each of the LLP-2 residues in the endogenous *env* locus, creating 10 Env libraries that each represent 64 alleles (every possible codon). Next, we infected SupT1 cells at low MOI (multiplicity of infection) with each library individually and measured the change in allele frequencies after approximately six generations of viral spread (*Figure 3A*). We then combined analyses of these libraries with previously generated deep mutational scanning data produced for the Rev NES from a rev-in-*nef* virus (*Fernandes et al., 2016*). These fitness landscapes (*Figure 3B and C*) represent the amino acid preference of each gene when the overlap is removed. We note that, as these DMS datasets were performed independently of one another and display different strengths of selection as demonstrated in the difference in stop codon fitness, comparisons between Rev and Env enrichment ratios require normalization. To this end we provide rescaled enrichment ratios as described in *Gray et al., 2017*, and plot both the standard enrichments and rescaled data (*Figure 3—figure supplement 1*).

The LLP-2 fitness landscape shows that while LLP-2 is generally highly mutable, it has a strong preference for a negatively charged side chain (D or E) at residue 773 (*Figure 3B*). Although the importance of positive charges in LLP-2 has been suggested previously (*Kalia et al., 2003*; *Steckbeck*

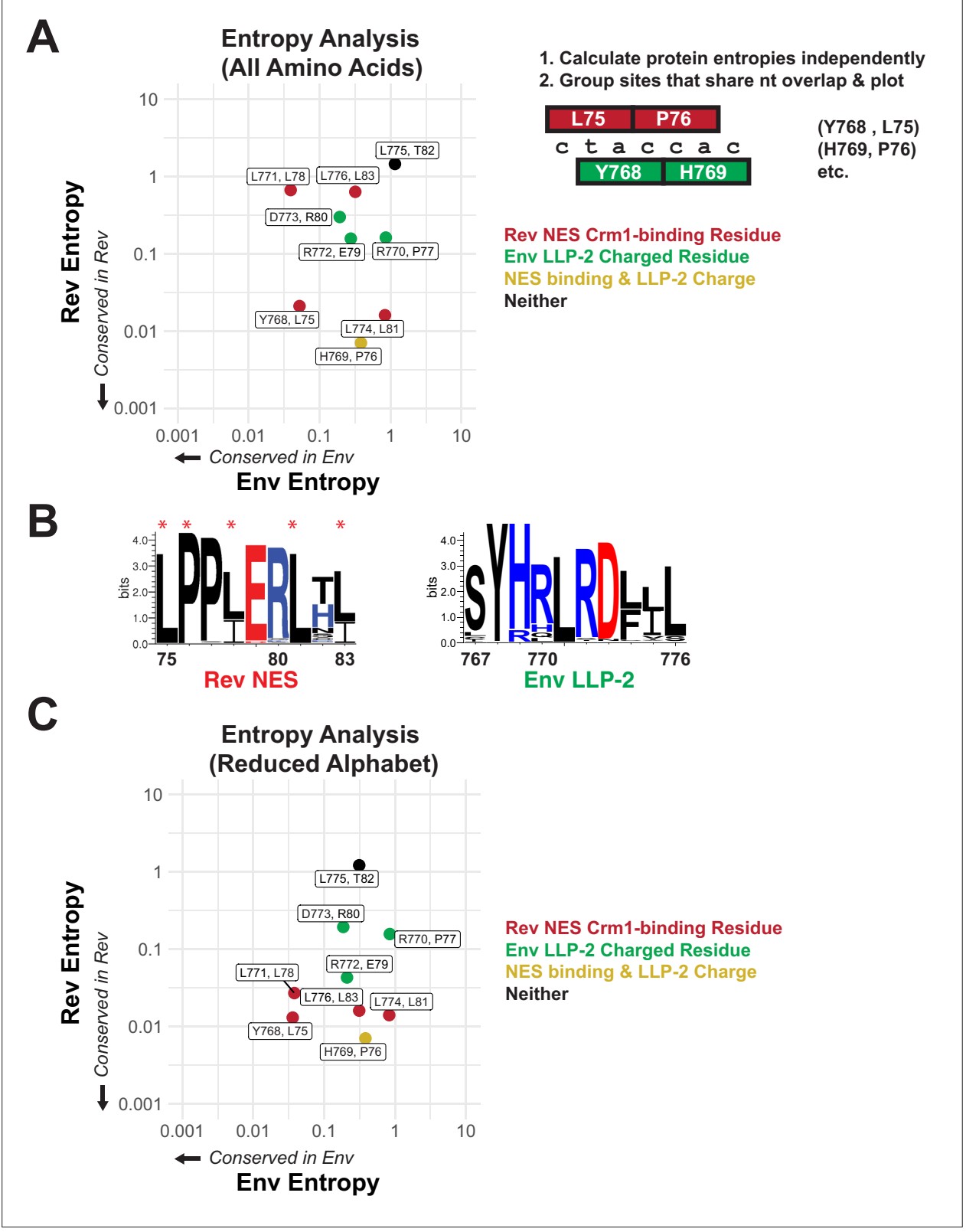

**Figure 2.** Rev nuclear export sequence (NES) and Env LLP-2 conservation. (**A**) Shannon entropies (low entropy = high conservation) from alignments of patient sequences (see panel B). Residues in Rev and Env are grouped by positions that share two nucleotides of overlap. Red points contain Crm1 contact residues, green points are charged LLP-2 residues, and yellow points contain both. (**B**) Sequence logos of Rev NES and Env LLP-2 showing sequence conservation at each position computed from a sequence alignment of patient data from the Los Alamos Database (https://www.hiv.lanl.gov/

*Figure 2 continued on next page*

*Figure 2 continued*

content/index; 2018). Positively charged residues (KRH) are in blue; negatively charged residues (DE) are in red. Functionally important residues of Rev are marked with a star. (**C**) Shannon entropies recomputed for a simplified amino acid alphabet (D=E, R=K, I=L and V, M, F=W & Y, N=Q, S=T). Coloring is as in 2A.

The online version of this article includes the following source data for figure 2:

**Source data 1.** Site-specific entropy values used to make graphs in *Figure 2*.

*et al., 2011*), our experiment is the first to identify strong selection for a negative charge. The data also show strong selection against the 'helix-breaker' proline (*Li et al., 1996*) at all positions, further underscoring the importance of its helical structure. Stop codons are robustly selected against as well, demonstrating the importance of the cytoplasmic tail in HIV-1 replication. Additionally, L771, which is surrounded by two positively charged arginines, displays a strong preference against charged residues. The Rev NES fitness landscape is consistent with many mutational studies, showing selection for large hydrophobic side chains (ILV) at each Crm1 contacting residue. The patient isolates do not show such a striking preference (*Figure 2B*), perhaps reflecting constraints imposed by overlapping gene mutational paths (e.g. any transitional mutation from L to V would have to be fit in Rev and also in Env) or the existence of other in vivo pressures not captured in our experiments (e.g. immune or cell-type-specific responses). Unlike LLP-2, we do not observe selection against proline in the semi-helical Rev NES, consistent with the presence of prolines in its reference sequence.

## Functional assays confirm the importance of Rev NES hydrophobic residues and LLP-2 charged residues

To confirm that our selection data reflect Rev functional requirements, we assayed individual alanine mutants at the nine NES residues in a p24 Rev-dependent RRE reporter assay (*Smith et al., 1990*). Mutation of any of the leucines (positions 75, 78, 81, and 83) or proline at position 76 resulted in loss of function (*Figure 4A*; high p24=active Rev), consistent with previous data of Crm1-binding residues (*Figure 1C*; *Fornerod et al., 1997*; *Güttler et al., 2010*; *Booth et al., 2014*; *Jayaraman et al., 2019*). Consistent with our selection data, mutation of the NES non-binding face had limited impact on Rev function.

To confirm that our selection data reflect LLP-2 functional requirements, we measured the infectivity of virions generated from pseudotyped *env*-deficient viruses complemented with Env LLP-2 alanine mutants in trans. The D773A mutation produced the most drastic phenotype (15% of wild-type Env), consistent with our selection data (*Figure 4B*). Previous work noted the conservation of arginines within the LLPs (*Kalia et al., 2005*; *Steckbeck et al., 2011*) and our data confirms that the first arginine in LLP-2 (R770) is also important for Env function, in addition to D773.

## Charged LLP-2 residues facilitate Env function in virus replication assays

As D773 in LLP-2 is clearly an important residue based on the selection and reporter data, we examined the effects of several substitutions at this position on viral replication. D773E, a conservative change predicted to maintain function based on our fitness data, showed ~50% of wild-type activity in the reporter assay, but substantially higher than the non-conservative D773A and D773N substitutions (*Figure 5A*). These results correlated well with the replication of rev-in-nef viruses engineered with the mutations (*Figure 5A*). It is interesting that D773A viruses still can replicate, albeit poorly, despite its very low activity; these results suggest that the virus can partially compensate for the loss of function, particularly when not competing with the wild-type sequence. The weakly active D773N allele is observed in ~2.6% of patient isolates (*Figure 2B*), further suggesting that the virus has alternative mechanisms to achieve the function normally performed by the negative charge at this position.

Given the importance of D773 and the neighboring positively charged R770 and R772 (*Figure 4B*), we sought to establish the importance of charge and spacing along the LLP-2 helix, with the goal of understanding what evolutionary constraints – absent the overlap – affect this region. We generated two double mutants with D773/R770 or D773/R772 changed to alanine (R1D→AA or R2D→AA) as well as the triple mutant (RRD→AAA) and found that mutation of R770 was particularly detrimental, in combination with D773 which is about one helical turn away (*Figure 5B*). Both of the double mutants

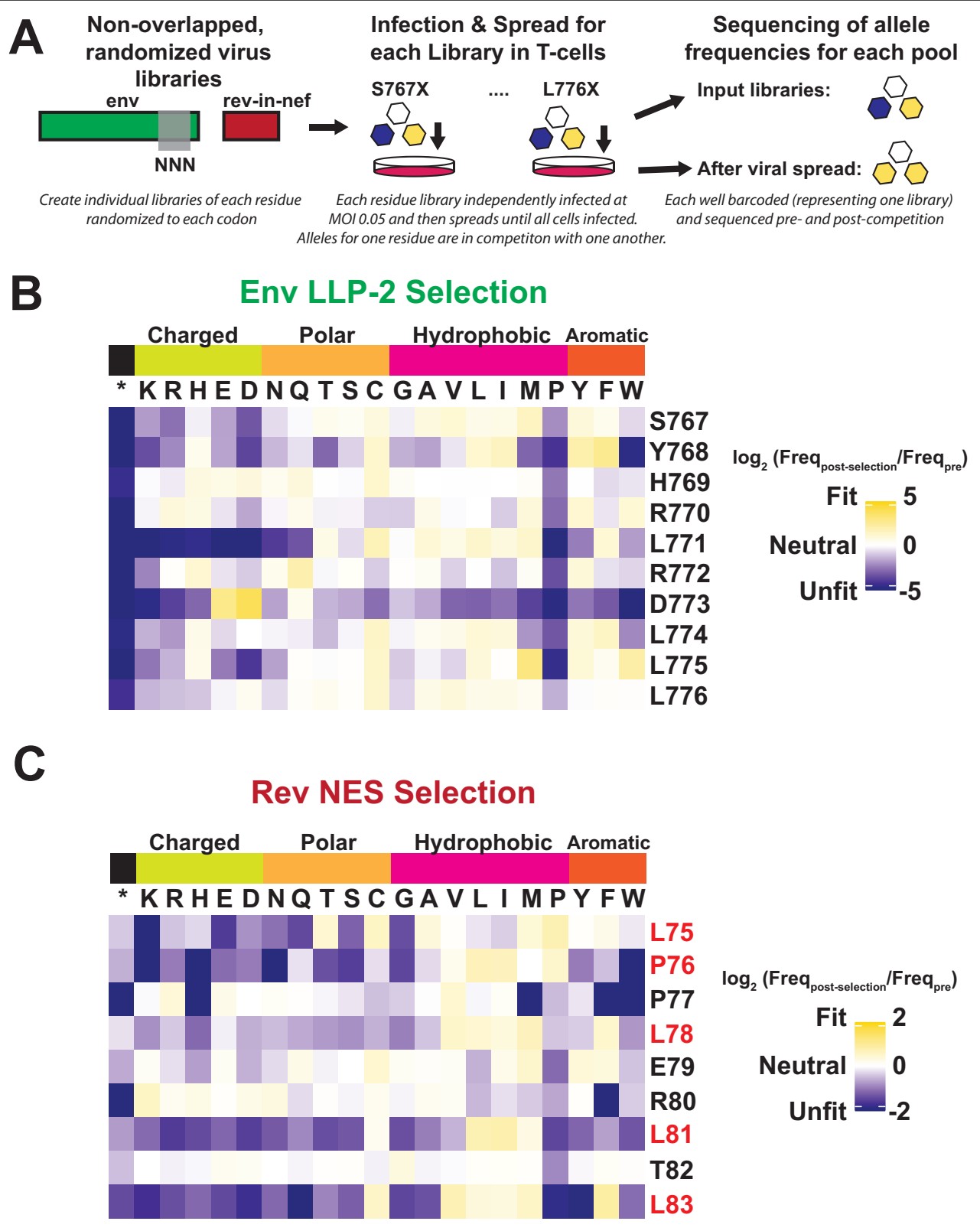

**Figure 3.** Deep mutational scanning of Rev nuclear export sequence (NES) and Env LLP-2. (**A**) Schematic of the experiment. Non-overlapped viruses with *rev* inserted in the *nef* locus were created and the endogenous LLP-2 region randomized to NNN to create 10 independent libraries. These libraries were then arrayed and allowed to spread (~6 generations of viral spread) and next-generation sequencing was used to determine frequencies of each variant in a replication competition experiment and to calculate experimental fitness. (**B**) Fitness (blue = unfit, white = neutral, and gold = fit) for every

*Figure 3 continued on next page*

*Figure 3 continued*

allele in the region of the Env LLP-2 that overlaps the Rev NES. Row headers denote the reference sequence while column headers specify the allele. (**C**) Fitness for every allele in the region of the Rev NES that overlaps the Env LLP-2. Functionally important residues of Rev are labeled in red.

The online version of this article includes the following source data and figure supplement(s) for figure 3:

**Source data 1.** Fold change values in allele frequency during deep mutational scanning experiment for Rev nuclear export sequence (NES) used to generate heatmaps.

**Source data 2.** Fold change values in allele frequency during deep mutational scanning experiment for Env LLP-2 used to generate heatmaps in *Figure 3*.

**Figure supplement 1.** Normalized Enrichment Scores for deep mutational scanning of Rev nuclear export sequence (NES) and Env LLP-2.

showed significantly decreased virus replication (*Figure 5B*), again suggesting loss of function can be partially compensated in the viral context for the loss of function. The triple mutant was highly defective in both reporter and replication assays indicating the importance of these charged residues. Western blots of these virions indicated decreased Env protein stability for R1D.

We constructed a set of additional charged residue mutants, including H769 as well as substitutions with lysine, to determine if their contributions could be attributed solely to charge. The results show no functional phenotype for H769K mutant in both reporter and replication assays and RR→KK outcomes confirm the importance of R770 and R773 (*Figure 5C*). While we did not examine the mutations of R770K and R773K independently out, deep mutational scanning of Env LLP-2 (*Figure 3B*) reveals selection against negative charged residues (D and E) and even K for both arginines, a bias is not observed in H769, suggesting that H769 is less constrained than the two arginines. It is interesting that substitution of both arginine with lysine is insufficient to maintain function or replication, and indeed the (RR→KK) mutant showed the most severe replication defect. It is possible that lysine substitutions might serve as a target for host restriction factors (*Zhang et al., 2021*) or introduce ubiquitination sites leading to protein turnover. Regardless of the mechanism, the data suggest that these particular residues are subject to significant evolutionary constraints. The results are also consistent with analyses of the Env cytoplasmic tail, which has shown enrichment for arginine and depletion for lysine when compared to the rest of Env and all other proteins found in the Uniprot database (*Steckbeck et al., 2011*).

## Helical requirement for LLP-2 function

LLP-2 forms two helices (*Murphy et al., 2017*) and our deep mutational scanning data suggest strong selection against helix-breaking prolines, implying a structural constraint during evolution. To better understand the extent of this constraint, we replaced residues 767–776 with glycines and serines (10GS) to create a flexible linker. As anticipated, the 10GS variant had no activity in the reporter assay and was unable to replicate even when D773 was reintroduced (*Figure 6A*). We next replaced residues 767–776 with alanines (10A) which are often used to form rigid helical linkers (*Chen et al., 2013*). This 10A variant was similarly defective (*Figure 6B*). However, in this case, reintroducing important charged residues (R770 or D773 or the five-residue HRLRD segment) resulted in substantial restoration of activity and rescue of virus replication, particularly in the cases where D773 was present (*Figure 6B*). These data further confirm the importance of charge and helical structure in LLP-2 function (*Kalia et al., 2005*; *Boscia et al., 2013*; *Steckbeck et al., 2014*).

## Functional segregation of residues in overlapping structured segments

Our experiments establish the functional restrictions in Env LLP-2 without competing selection pressure from Rev, permitting us to map the constraints of each reading frame onto the nucleotide sequence of the virus (*Figure 7*). Specifically, our analysis showed that in Env, D773 has a strong preference for negative charge, R770 and R772 have an intermediate preference for arginine side chains, H769 has a weak preference for a positive charge, and L771 selects against charged side chains. When combined with the preferences of the Rev NES Crm1-binding residues (*Fernandes et al., 2016*), it is apparent that the functional surfaces of each domain are positioned so as to minimize negative consequences of the overlap. The non-binding face of the Rev NES largely corresponds to the charged surface of the Env LLP-2 and the hydrophobic surface of the LLP-2 corresponds to the binding residues of the Rev NES (*Figure 7*). The majority of overlaps between opposing critical residues mostly involve mutable

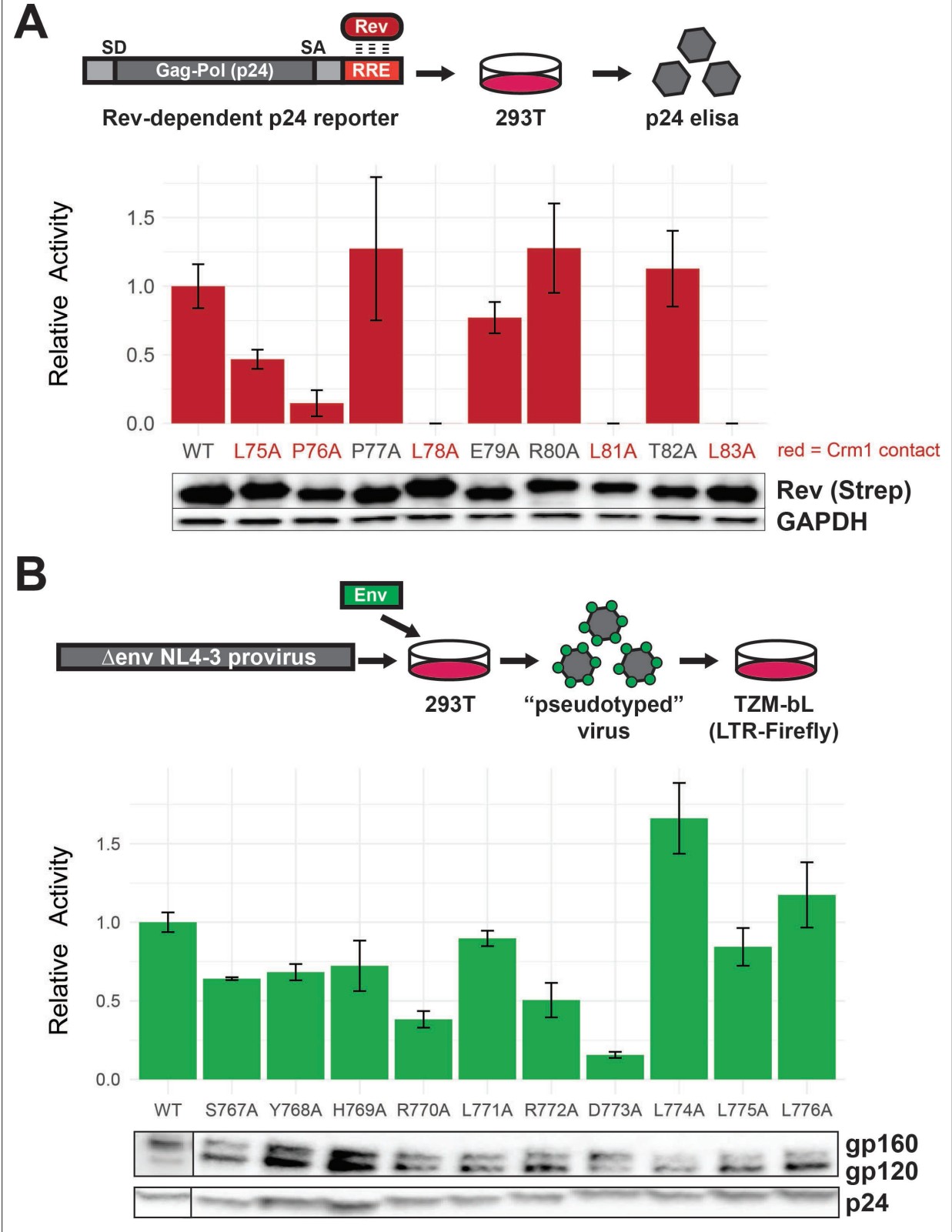

**Figure 4.** Functional dissection of Rev nuclear export sequence (NES) and Env LLPs. (**A**) Schematic of the Rev reporter assay and activities of individual NES alanine mutants. To monitor Rev export activity, an RRE-containing gag-pol reporter system that produces viral capsid (**p24**) in a Rev-dependent manner was used. Data are mean ± standard deviation (s.d.) of three biological replicates. Western blot shows expression levels of Strep-tagged Rev variants and GAPDH as loading controls. Functionally important residues of Rev are labeled in red. (**B**) Schematic of the Env reporter assay in the

*Figure 4 continued on next page*

*Figure 4 continued*

TZM-bL system and results with individual Env LLP-2 alanine mutants. Data are mean ± s.d. of three biological replicates. Western blots show Env incorporation (gp120 antibody) into pseudoviruses and p24 as loading controls. For these experiments, we ablated the *env* reading frame by mutating the start codon and introducing multiple downstream stop codons. Plasmids were co-transfected into 293T cells with each alanine mutant to generate Env pseudotyped viruses. The resulting virions were used to infect TZM-bL reporter cells which express the HIV-1 surface receptors CD4, CCR5, CXCR4, and produce luciferase under the control of the *tat*-dependent HIV-1 LTR (*Sarzotti-Kelsoe et al., 2014*). A functional Env protein permits binding and entry and expression of Tat and thus reflects the number of infectious viral particles in a single round of infection, but the integrated virus lacks the *env* gene and cannot initiate additional rounds of replication. Incorporation of Env mutants into virions was also measured by Western blot.

The online version of this article includes the following source data for figure 4:

**Source data 1.** Reporter data used to generate graphs for Rev and Env alanine scans in *Figure 4*.

**Source data 2.** Unedited Western blot for *Figure 4*.

wobble bases, with the sole exception being the Env H769/Rev P76 overlap. However, in this case Rev appears to be the major driver of evolution as mutation of P76 results in a severe loss of function whereas H769 appears to play a minor, supporting role in the function of the charged surface of LLP-2. Env L771 also overlaps Rev L78, but contributes relatively minor constraints at the nucleotide level, as its selection against polar residues, rather than for a particular side chain, permits multiple codons with high fitness in LLP-2.

## LLP-2/NES overlaps in other lentiviruses

The segregation of functional residues in LLP-2 and NES prompted us to examine the Env/Rev overlap among primate lentiviruses. The Env cytoplasmic tail is moderately conserved between HIV-1 clades and the extent of their sequence variation is similar in regions that do or do not overlap with Rev (*Steckbeck et al., 2011*). This observation is consistent with our finding of functional segregation, where the overlapping region essentially acts like two single reading frames. The four distinct HIV-1 lineage groups (M, N, O, P) share similarity with other primate lentiviruses that originated from two different simian immunodeficiency viruses (SIV) and all have overlapping LLP-2/NES regions (*Figure 8*).

To explore these regions in depth, we extracted Rev NES and Env LLP-2 sequences from curated amino acid alignments of HIV-1/SIVcpz, HIV-2/SIVsmm, and other SIVs in the Los Alamos Database (2018; hiv.lanl.org) and compared sequence logos for each virus (*Crooks et al., 2004*; *Foley, 2013*; *Figure 8*). The sequence alignment and number of sequences used for each sequence logo is provided in *Supplementary file 1*, *Supplementary file 2*. The hydrophobic Crm1-binding residues of the NES are highly conserved across different HIV-1, HIV-2, and SIVs (*Figure 8*) but the NESs can be broadly grouped into two categories: one contains the hydrophobic/proline arrangement observed in HIV-1 and the other contains HIV-2-like characteristics that preserve the spaced hydrophobic contacts but lack consecutive prolines and instead are glutamine-rich. LLP-2 is more varied and viruses fall into three groupings: (1) HIV-1 M, N, and SIVcpz*Ptt* conserve the charge cluster dissected in our assays; (2) HIV-1 O, HIV-1 P, SIVgor, and SIVcpz*Pts* have minimal charge; and (3) SIVmac, HIV-2, SIVsmm, and SIVrcm conserve R772 and occasionally D773. The high conservation of hydrophobic Crm1 contacts appears to dominate selection for the NES while maintaining sufficient sequence flexibility for LLP-2 to retain its structure. Indeed, the size of the Env cytoplasmic tail varies among different retroviruses (*Tedbury and Freed, 2015*), and the loss of the tail is commonly observed when SIV is propagated in human cells, suggesting that both host and viral determinants influence the function and variability of this region (*Kodama et al., 1989*; *Shimizu et al., 1992*; *Shacklett et al., 2000*).

## Discussion

We previously found that overlapped genes (*rev* and *tat*) can segregate functionally important residues by having a disordered segment of one protein overlapped with an ordered segment of the other (*Fernandes et al., 2016*). The present study shows that residues can functionally segregate even when both overlapping segments are structured. We probed residues in the Env gp41 LLP-2 that overlap with the Rev NES and compared the limits to which this domain can tolerate mutation to the already well-established mutational limits of Rev (*Jayaraman et al., 2019*). Interestingly, we observed that critical residues in both proteins are located on one face of a helix such that they are spaced periodically in alternative reading frames (*Figure 7*). This arrangement is a consequence of

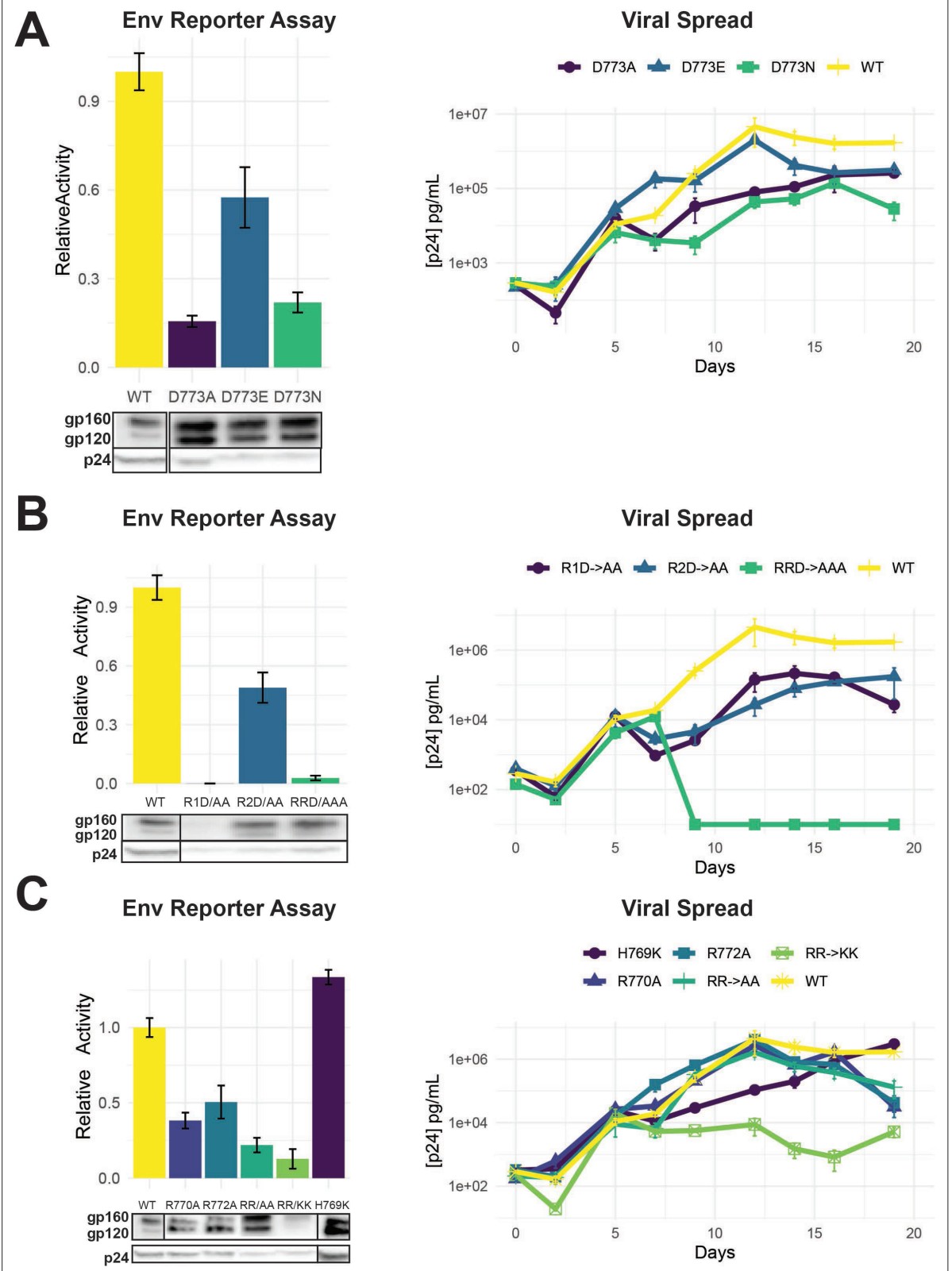

**Figure 5.** Importance of LLP-2 charged residues for replication. (**A**) Left: Env functional assays as described in *Figure 4* with different D773 mutations. Right: Viral replication kinetics in spreading assays using NL4-3 rev-in-nef proviral plasmids (*Fernandes et al., 2016*) engineered with Env D773 mutations. (**B**) Left: Env functional assays showing the role of three charged residues. Western blots show incorporation of Env mutants in virions (dark boxes represent a break in the lane loading presentation, with all samples loaded on a single gel). Right: Viral replication kinetics of viruses engineered

*Figure 5 continued on next page*

*Figure 5 continued*

with the mutations. R1D→ AA is a double mutant with R770/D773 changed to alanine, R2D→ AA is a double mutant with R772/D773 changed to alanine, and RRD→ AAA a triple mutant R770/R772/D773 changed to alanine. (C) Left: Env functional assays showing effects of positive charge mutations with corresponding Western blots of Env incorporation in virions. Right: Viral replication kinetics of viruses engineered with the mutations. Replication assays are represented on a log scale. All data are mean ± standard deviation (s.d.) of biological triplicates.

The online version of this article includes the following source data for figure 5:

**Source data 1.** Reporter data for Env mutants in *Figures 5 and 6*.

**Source data 2.** Viral spread data used to generate graphs for Env mutants in *Figures 5 and 6*.

**Source data 3.** Unedited Western blot for *Figure 5*.

the genetic code where the wobble bases of functionally critical codons in one reading frame can be chosen to accommodate critical residues in the alternative frame, while the first two codon bases overlap with more mutation-tolerant residues located on a non-binding surface of the overlapped protein (*Figure 7*). Specifically, the important hydrophobic residues of the Rev NES are oriented on one surface toward the Crm1-binding groove, with consecutive prolines altering the helical register compared to other Crm1 cargoes (*Booth et al., 2014*; *Jayaraman et al., 2019*). The non-binding face of the NES helix is highly mutable and capable of sampling nearly every amino acid with roughly equal fitness. These highly mutable residues correspond to nucleotides that encode critical charged residues on the functional surface of gp41 LLP-2 in the alternative frame. Correspondingly, residues on the non-charged face of the LLP-2 helix minimally constrain codon usage of important Rev NES contact residues.

The striking segregation of function we observe for Env LLP-2/Rev NES is made possible by the atypical proline-rich NES and partially redundant function of the LLP-2 positive charges. This manner of segregation in HIV-1 is not necessarily the case for similarly overlapped regions in other lentiviruses, which appear to have different functional requirements based on their sequence conservation (*Figure 8*). The findings are another indication of how efficiently HIV-1 makes use of its limited coding capacity.

Our results show that Env D773 is under strong selective pressure although the precise function of this residue is not known. In addition to this negative charge, positive charges positioned on the same face of the helix are also important. Previous studies have suggested several possible functions for the LLP-2, including maintaining Env in a prefusogenic state (*Lu et al., 2008*; *Postler and Desrosiers, 2013*; *Murphy et al., 2017*), regulating viral assembly and egress (*Snetkov et al., 2021*), or mediating matrix-Env interactions (*Alfadhli et al., 2019*). The function of these charged residues in viral fusion or assembly remains to be determined.

We note that even in our model of functional segregation, two sites in Env overlapping portions of the Rev NES (Y768; overlapping Rev L75 and P76, and L771; overlapping Rev L78) do have mild amino acid preferences (Y768 for large hydrophobic/aromatic residues and L771 for non-polar residues). In our model, in which we weigh both the selection and reporter data together, we view these mild constraints as selection against disruptive interactions or sites which contribute (but are not essential to) overall protein stability. However, it is important to note that this assumption is based on data from a single genetic background and in other genomic contexts or selection backgrounds these mild selective pressures may in fact be more stringent (*Canale et al., 2018*).

HIV-1 has evolved in an elegant manner to balance the functions of overlapped proteins, however the strategy may also expose a weakness whereby immune targeting of the LLP-2 could provide a fruitful therapeutic route. The LLP-2 is transiently surface exposed on the cell membrane during infection (*Kalia et al., 2005*; *Lu et al., 2008*; *Murphy et al., 2017*), and the cytoplasmic tail can modulate the antigenic properties of Env as a whole (*Piai et al., 2020*). Targeting of the LLP-2 surface with combinations of rationally designed immunotherapies may force the virus to mutate in ways that break the segregation between Env and Rev. Although our analysis of other lentiviruses suggests that HIV-1 might evolve other ways to establish Rev function under such pressure, it is likely that escape would still require multiple mutations and may be less likely in combination with other therapies. Deep mutational scanning has proven effective at mapping probable mutational escape pathways from immune pressures such as neutralizing antibodies (*Starr et al., 2020*; *Greaney et al., 2021*) and the Env cytoplasmic tail has proven to be a viable target in primate animal models *Wang et al., 2020*; for

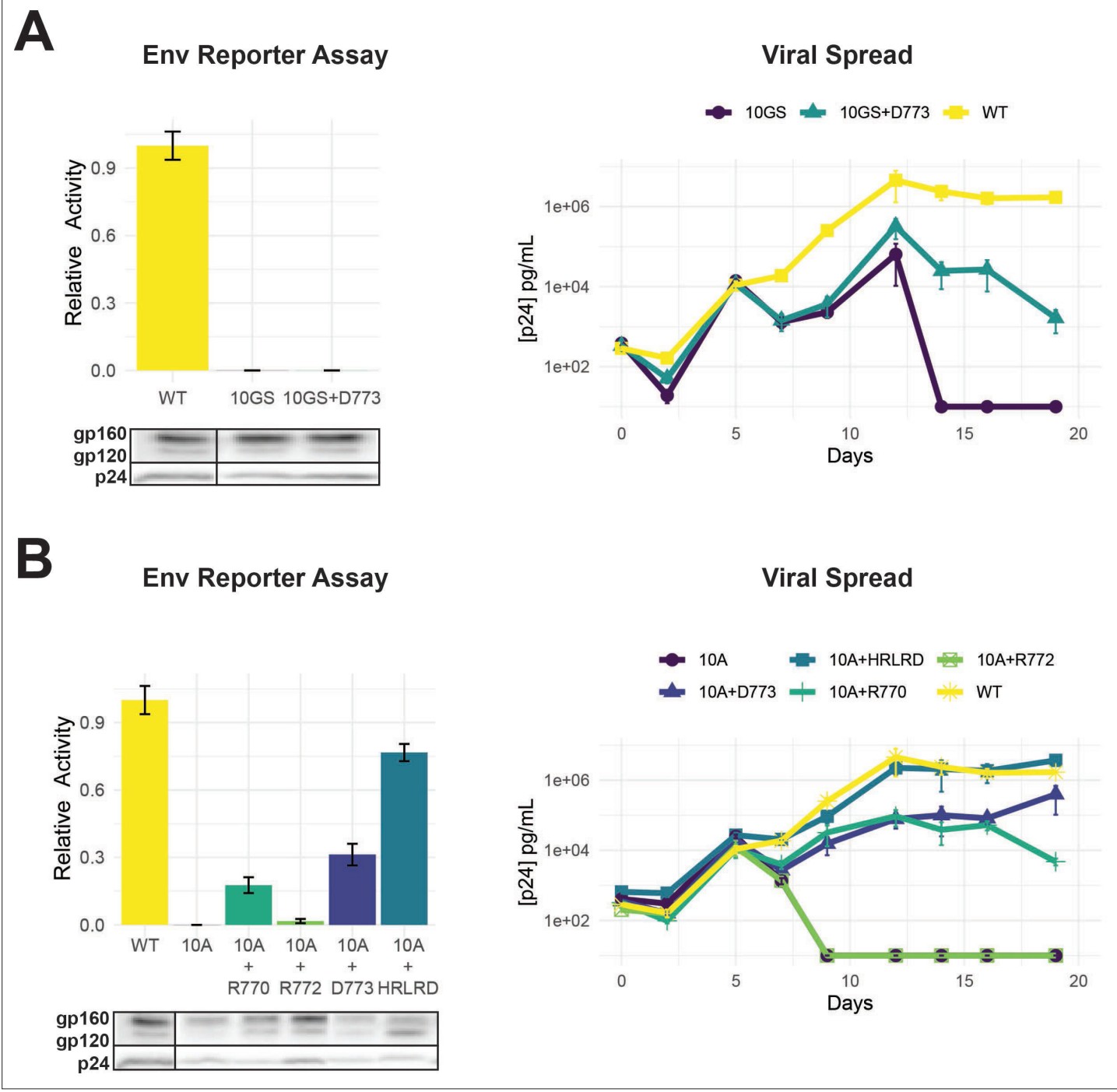

**Figure 6.** Importance of LLP-2 structure and charge for Env activity. (**A**) Left: Env functional assays as described in *Figure 4* when LLP-2 helical structure was disrupted by replacing 10 residues (767–776) with alternating glycine/serine residues (10GS) to create a flexible region. Western blots show expression of Env mutants (dark boxes represent a break in the lane loading presentation, with all samples loaded on a single gel). Right: Viral replication kinetics of wild-type (WT), 10GS, and 10GS + D262 viruses. (**B**) Left: Env functional assays with charge mutants within a 10-alanine (10A) context designed to maintain a helical structure with corresponding Western blots of Env incorporation in virions. Right: Viral replication kinetics with helical structure mutant viruses. Data points in both panels are mean ± s.d. of biological triplicates. Replication assays are represented on a log scale.

The online version of this article includes the following source data for figure 6:

**Source data 1.** Unedited Western blot for *Figure 6*.

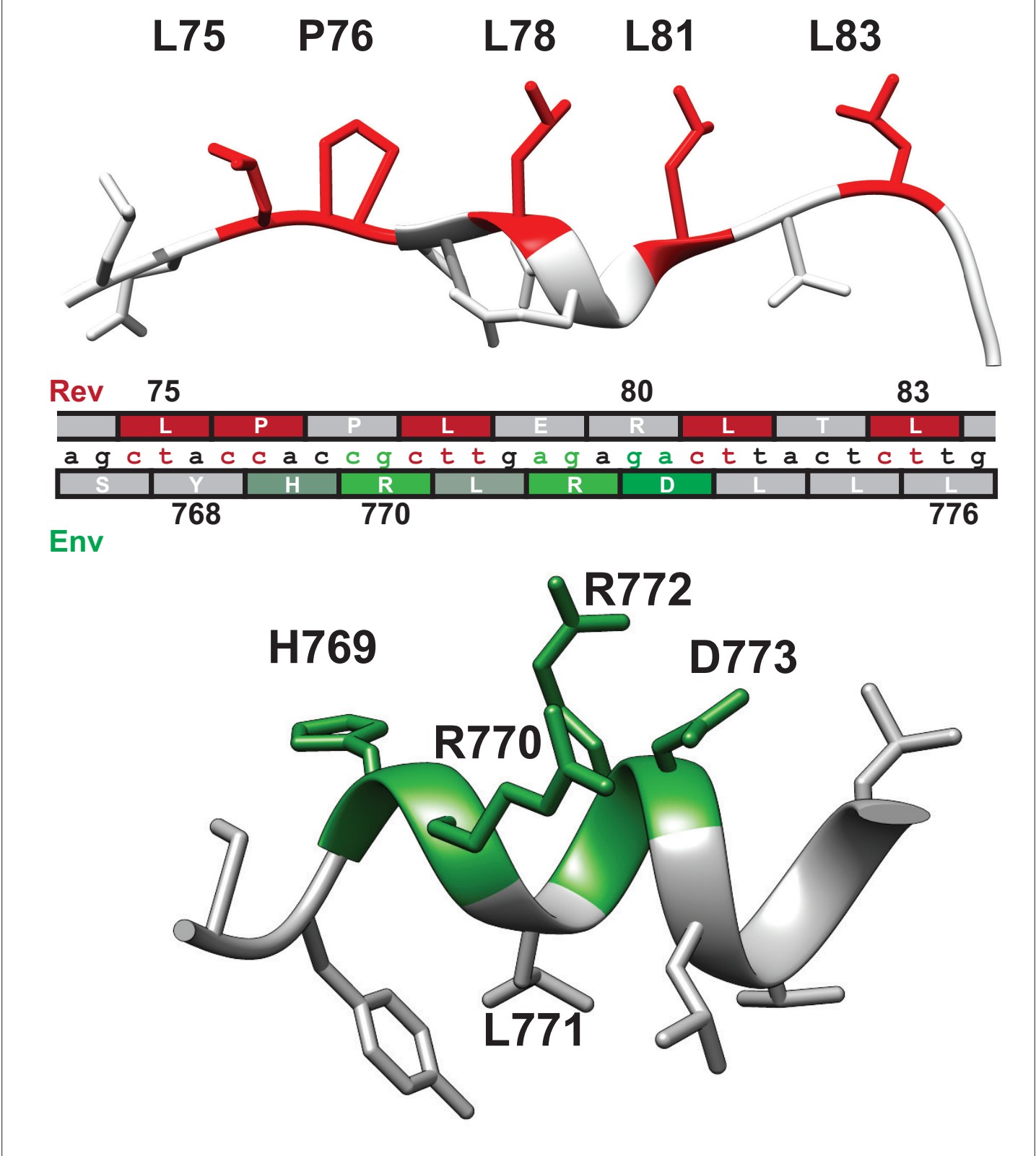

**Figure 7.** Nucleotide segregation and protein structure of overlapping Rev nuclear export sequence (NES) and Env LLP-2. Functionally important residues of the Rev NES (red, CRM1-binding residues) and Env LLP-2 (green, charged residues) are highlighted in their respective structures (PDB: 3nc0 and PDB: 6ujv) and shown with the corresponding nucleotide sequence.

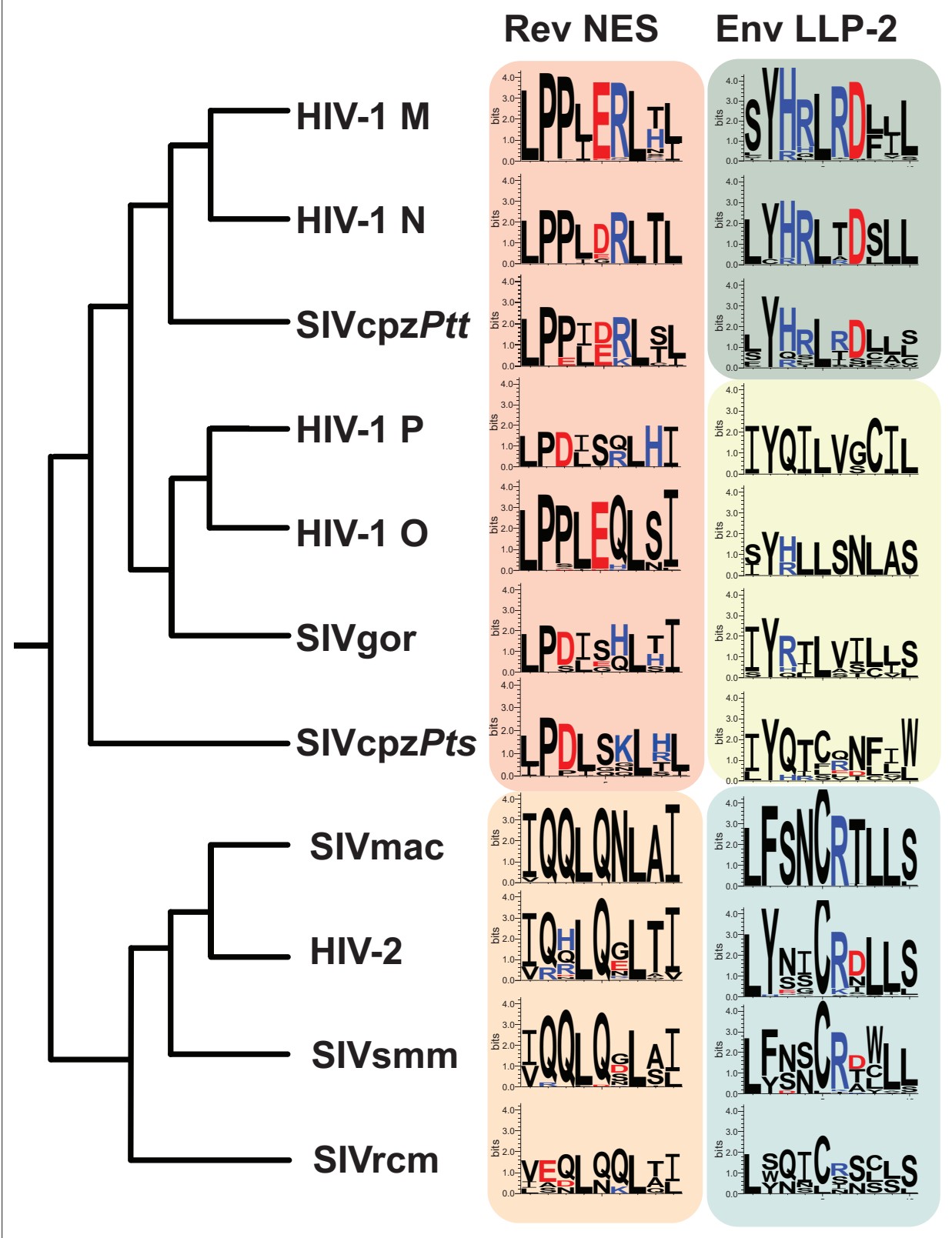

**Figure 8.** LLP-2/NES overlaps in primate lentiviruses reveals variation of NES and LLP-2. Phylogenetic relationship of different primate lentiviruses, including four HIV-1 groups (**M, N, O, P**), SIVcpz*Pts* (from eastern chimpazees), SIVgor (from gorillas), SIVcpz*Ptt* (from central chimpanzees), HIV-2, SIVsmm (from sooty mangabeys), SIVmac (from macaques), and SIVrcm (from red-capped mangabey). Sequence logos showing conservation of the corresponding Rev NES and Env LLP-2 regions of each virus computed from their sequence alignments (extracted from 2018 curated amino acid

*Figure 8 continued on next page*

*Figure 8 continued*

alignments of HIV-1/SIVcpz, HIV-2/SIVsmm, and other SIV from the Los Alamos Database (hiv.lanl.org)). Positively charged residues (KRH) are indicated in blue and negatively charged residues (DE) in red. NES, nuclear export sequence.

The online version of this article includes the following source data for figure 8:

**Source data 1.** Number of sequences used to create each sequence logo in *Figure 8* in addition to each sequence alignment.

HIV-1 the combined knowledge of both functional constraints in the proteins and coding constraints in the genome may allow targeted immune therapies that drastically limit (or prevent altogether) escape mutations. Indeed, immune targeting of overlapped reading frames in in vivo models has been shown to drive predictable synonymous mutations in the alternative, non-targeted frame (*Hughes et al., 2001*). Similar approaches in this region of dual structure and function may provide a novel way of limiting HIV's otherwise unpredictable mutational escape patterns.

# Materials and methods

## Cells

HEK 293 (obtained from ATCC) and TZM-bL cells (obtained from the NIH AIDS Reagent Program) were maintained in humidified incubators at 37°C and 5% $CO_2$ in DMEM, high glucose (with 10% heat-inactivated fetal bovine serum [FBS], and 1% penicillin/streptomycin). SupT1 human T-cell lines (obtained from ATCC) were maintained in RPMI (supplemented with 25 mM HEPES pH 7.4, 10% heat-inactivated FBS, and 1% penicillin/streptomycin) at 37°C and 5% $CO_2$. All cell lines had their identity confirmed by STR profiling at the UC Berkeley Cell Culture facility and tested negative for mycoplasma.

## Plasmids

Different mutants were constructed using primers (synthesized by IDT) harboring the appropriate mutations, and then used in a two-step PCR and inserted into pcDNA4/TO-2xStrep vector (described in *Fernandes et al., 2016*) by Gibson Assembly for testing in export assays. Viral mutants were created by the same strategy in pNL4-3 rev-in-nef construct (*Fernandes et al., 2016*).

## Rev and Env reporter assays

Nine residues of Rev and 10 residues of Env, overlapping region of Rev NES and Env LLP-2, were mutated to alanine in independent site directed mutagenesis reactions in mammalian expression vectors. For the Rev reporter assays, 293T cells in a 24-well plate were transfected with 250 ng of pCMV gagPol-RRE and 25 ng of a pcDNA4/TO 2xStrep NL4-3 Rev using PolyJet (SignaGen) reagent. Cells were lysed in Tris-buffered saline containing 1% Triton X-100 and protease inhibitors, 48 hr after transfection, and intracellular p24 levels were measured by ELISA. The assay was performed in biological triplicate. For all mutants, activity was normalized to the unmutated reference activity. Rev expression levels were verified by Western blot using HRP-conjugated anti-StrepTagII antibody (IBA Biosciences) with GAPDH used as a loading control.

For Env reporter assays, the relative infectivity of Env pseudoviruses of LLP-2 mutants was measured in TZM-bl cells assay. To prepare HIV-1 Env pseudoviruses of LLP-2 mutants, 293T cells were co-transfected with pNL4-3 ΔEnv and an envelope expressing plasmid pcDNA4/TO Env. Pseudovirus containing culture supernatant was harvested 48 hr after transfection and concentrated by PEG purification (the final concentrations for PEG-6000 and NaCl were 8.5% (w/v) and 0.3 M, respectively) (*Kutner et al., 2009*). The infectivity of mutants was determined by infecting TZM-bl cells with p24-normalized pseudoviruses. Forty-eight hours post-infection, luciferase activity of the reporter gene was quantified using Bright-Glo luciferase reagent (Promega, Madison, WI). Experiments were done in biological triplicate and relative infectivity was normalized to the unmutated reference infectivity. The incorporation of Env LLP-2 mutants into the pseudovirus was measured by Western blot using gp120 antibody and p24 as loading control.

## Viral replication assays

NL4-3 rev-in-nef proviral constructs (*Fernandes et al., 2016*) were used to introduce mutations in the Rev NES or Env LLP-2 in non-overlapped contexts. For generation of each virus, HEK 293T cells in six-well plates were transfected with 1000 ng of NL4-3 proviral plasmid and complemented with reference Rev or Env using PolyJet (SignaGen) reagent at a 1:3 ratio of the DNA:Polyjet. Viral supernatants were harvested 48 hr after transfection, stored at −80°C, and then virus titer quantified by p24 capsid ELISA. One million SupT1 cells were incubated with 5 ng of p24 virus in 250 µl media containing 1 µg/ml PEI and 8 µg/ml polybrene and centrifuged 1200 $g$ for 2 hr at 37°C. Input virus was removed and then washed cells with 250 µl of PBS. Cells were resuspended in 250 µl of media and placed in an incubator at 37°C, 5% $CO_2$. Every 48–72 hr for 20 days, 100 µl of virus-containing supernatants were collected for quantification by p24 ELISA and added 100 µl of fresh media to cells. Viral p24 levels below 10 pg/ml are shown as 10 pg/ml in the plots.

## Competitive deep mutational scanning analysis

The Rev NES fitness values were extracted from previous work (*Fernandes et al., 2016*). Env LLP-2 proviral libraries were created as described (*Fernandes et al., 2016*). Briefly, each of the 10 individual residues of Env LLP-2 that overlap with the Rev NES was randomized. For this purpose, this region was amplified using 10 separate pools of 33 nt primers containing NNN in the middle of the primer and using Gibson Assembly Master Mix (NEB) was cloned into pNL4-3 rev-in-nef after digesting by BamHI and XhoI. The proviral library of each residue was generated by scraping together at least 1000 colonies of each primer pool for plasmid purification. High codon diversity of each randomized proviral library (10 total) was confirmed by deep sequencing. Virus generation for each proviral plasmid library was done as described above in 293T cells. Selection assays were performed via spinoculation of one million SupT1 cells as above. The infections were examined by IFA for cellular HIV antigen synthesis and also by quantification of progeny virion release into supernatant using p24 ELISA. Once the infection reference or defective viral pools reach their peak, 150 µL of viral supernatant was collected and centrifuged to remove cells, and then RNA was extracted from virions by ZR-96 Viral RNA extraction kits (Zymo) and eluted in 15 µl water. cDNA was amplified from 5 µL of extracted RNA using the iScript Advanced (BioRad) cDNA synthesis. The selection experiment was performed in biological duplicate for each pool/residue. Samples were sequenced on an Illumina MiSeq with PE100 and demultiplexed on instrument. Raw fastqs were aligned to the reference sequence with bowtie2 and the flags `--fast-local --rdg 100,3 --rfg 100,3`. Codon and amino acid counts were generated via the countDMS program available on GitHub. These counts were then log2 normalized by fractions calculated from sequencing the input proviral plasmids to generate experimental fitness scores. The resulting fitness matrices were then processed and plotted using the ComplexHeatmap package in R.

## Entropy

Entropy values were calculated using hiv.lanl.gov 'Entropy One' tool and the 2018 filtered web alignments (amino acid) for Rev and Env, discounting ambiguous characters. The 'amino acid equivalents' setting was used to analyze chemically conservative changes with the addition that I, L, and V were all treated equivalently.

## LLP-2/NES overlap in primate lentiviruses

The 2018 curated amino acid alignments of HIV-1/SIVcpz, HIV-2/SIVsmm, and other SIV were downloaded from the Los Alamos Database (hiv.lanl.org). The corresponding Rev NES and Env LLP-2 region of each virus was extracted and modified as needed in Geneious Prime v2019.1.1. The sequence conservation for each domain of each virus was generated using WebLogo3.

## Data and code availability

Requests for additional data or alternative formats should be addressed to the lead contacts. Sequencing data has been deposited in GEO: accession GSE179046; code is available on https://github.com/jferna10/EnvPaper, (copy archived at swh:1:rev:152ada7da67b08f3e04ac95b284e45999c90341c; *Fernandas, 2021*) All graphs were generated in Rstudio with associated files available on GitHub.

## Acknowledgements

We thank all members of the Frankel lab for helpful comments and review of the manuscript and members of the John Gross lab for beneficial discussion. We also thank Henni Zommer for making some of the plasmids containing different Env LLP-2 mutations. This work was supported by NIH grant P50AI150476, and MS was supported in part by NIH training grant T32AI0605357.

## Additional information

### Competing interests

Bhargavi Jayaraman: is currently an employee at Synthekine. The author has no financial interests to declare regarding this work. Jason D Fernandes: is an employee of Scribe Therapeutics. The author has no financial interests to declare regarding this work. The other authors declare that no competing interests exist.

### Funding

| Funder | Grant reference number | Author |
|---|---|---|
| National Institute of Allergy and Infectious Diseases | P50AI150476 | Alan D Frankel |
| National Institute of General Medical Sciences | T32AI0605357 | Maliheh Safari |

The funders had no role in study design, data collection and interpretation, or the decision to submit the work for publication.

### Author contributions

Maliheh Safari, Conceptualization, Data curation, Formal analysis, Investigation, Methodology, Writing – original draft, Writing – review and editing; Bhargavi Jayaraman, Conceptualization, Investigation, Writing – original draft, Writing – review and editing; Henni Zommer, Shumin Yang, Cynthia Smith, Data curation, Investigation; Jason D Fernandes, Conceptualization, Data curation, Formal analysis, Investigation, Methodology, Supervision, Writing – original draft, Writing – review and editing; Alan D Frankel, Conceptualization, Funding acquisition, Supervision, Writing – original draft, Writing – review and editing

### Author ORCIDs

Maliheh Safari ⬤ http://orcid.org/0000-0002-9511-6481
Bhargavi Jayaraman ⬤ http://orcid.org/0000-0002-5071-6117
Jason D Fernandes ⬤ https://orcid.org/0000-0002-8625-1796
Alan D Frankel ⬤ https://orcid.org/0000-0002-2525-9508

### Decision letter and Author response

Decision letter https://doi.org/10.7554/eLife.72482.sa1
Author response https://doi.org/10.7554/eLife.72482.sa2

## Additional files

### Supplementary files

Supplementary file 1. Fasta alignment of Env LLP-2 sequences used for analysis in *Figure 8*.

Supplementary file 2. Fasta alignment of Rev nuclear export sequence (NES) sequences used for analysis in *Figure 8*.

Transparent reporting form

### Data availability

Sequencing data have been deposited in GEO under accession codes GSE179046 Code is available on https://github.com/jferna10/EnvPaper, (copy archived at

swh:1:rev:152ada7da67b08f3e04ac95b284e45999c90341c). All graphs were generated in Rstudio with associated files available on github and/or in Supplementary files.

The following datasets were generated:

| Author(s) | Year | Dataset title | Dataset URL | Database and Identifier |
|---|---|---|---|---|
| Fernandes J | 2021 | Functional and Structural Segregation of Overlapping Helices in HIV-1 | https://www.ncbi.nlm.nih.gov/geo/query/acc.cgi?acc=GSE179046 | NCBI Gene Expression Omnibus, GSE179046 |
| Fernandes J | 2021 | Functional and Structural Segregation of Overlapping Helices in HIV-1 | https://github.com/jferna10/EnvPaper | GitHub, jferna10/EnvPaper |

The following previously published dataset was used:

| Author(s) | Year | Dataset title | Dataset URL | Database and Identifier |
|---|---|---|---|---|
| Fernandes J | 2016 | Deep Mutational Scanning of HIV tat and rev in a non-overlapped context | https://www.ebi.ac.uk/arrayexpress/experiments/E-MTAB-5154/ | ArrayExpress, E-MTAB-5154 |

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
