## [Editor Report]

Fernandes et al., ask the question: "What are the evolutionary constraints on genomic sequence that encode two different proteins?" To this end, they compare the functional constraints on mutations in HIV Rev and Env, which are encoded in different reading frames from the same region of the viral genome. Interestingly, residues that are functionally constrained in one protein are, for the most part, not as constrained in the other. The elegance of this solution is attractive and will be of interest to the protein evolution and structure communities.

---

## [Decision Letter]

**Decision letter after peer review:**

Thank you for submitting your article "Functional and Structural Segregation of Overlapping Helices in HIV-1" for consideration by *eLife*. Your article has been reviewed by 2 peer reviewers, and the evaluation has been overseen by a Reviewing Editor and Detlef Weigel as the Senior Editor. The following individual involved in review of your submission has agreed to reveal their identity: Marie-Louise Hammarskjöld (Reviewer #2).

Essential revisions:

1. Please see the two points in the Public Review.

2. The weak selection against stop codons for HIV Rev (Figure 3C) suggests that selection was weaker in that experiment. Because the authors compare the Env and Rev DMS datasets, they should provide rescaled data (potentially as a figure supplement), as described in Haddox et al., 2018.

3. The authors should clarify how many patient sequences were compared and whether isolates from all HIV-1 subtypes were included. It would be of interest to see a comparison between subtypes in the analysis.

4. To better support the statement that amino acid preferences are different in patients than in the deep mutational scan (page 9), the authors should calculate an entropy score from the DMS data and plot it against the patient entropy scores.

---

## [Author Response]

The weak selection against stop codons for HIV Rev (Figure 3C) suggests that selection was weaker in that experiment. Because the authors compare the Env and Rev DMS datasets, they should provide rescaled data (potentially as a figure supplement), as described in Haddox et al., 2018.

We agree with the reviewers’ concern that comparing across selection datasets requires some form of normalization for fairness. We have rescaled the data using the heuristic described in Grey et al., 2017 which was used to compare different DMS datasets and provide those factors as part of the supplemental tables and figures in support of Figure 3 (Gray, Hause and Fowler, 2017). While the rescaling model provided in Haddox et al., 2018 is appealing, we found it more difficult to implement and were unsure of the appropriateness of its assumptions in this case where the background genotypes are identical and the genes different (as opposed to DMS of the same gene in different genetic backgrounds) (Haddox *et al.*, 2018). We note that the Grey et al. 2017 metric is described as “roughly equivalent” in Haddox et al., 2018, and specify these caveats in the text.

**Author response image 1. sa2fig1:** 

3. The authors should clarify how many patient sequences were compared and whether isolates from all HIV-1 subtypes were included. It would be of interest to see a comparison between subtypes in the analysis.

We used alignments of 4113 Rev sequences and 7065 Env sequences, from all subtypes of HIV-1. Complete alignments have been made available as part of the supplementary data (HIV1_ALL_2018_env_LLP2 extraction.fasta and HIV1_ALL_2018_rev_NES extraction.fasta, and Figure 8 Supplementary Data S7)

Sequence conservation of different HIV-1 subtypes for the region of overlap is also shown in figure 8. The sequence logo of just HIV-1 group M (Figure 8) is similar to HIV-1 including different groups (Figure 2B), due to the low number of sequences available for other HIV-1 groups in comparison with HIV-1 group M. The number of sequences used for each sequence logo is figure 8 is listed in Author response table 1, which they are the only sequences available for each virus. In addition, the sequence alignment and number of sequences used for each sequence logo is provided in Figure 8 Source Data File S7.

**Author response table 1. sa2table1:** 

	Rev NES	Env LLP-2
HIV-1 M	4,054	7,004
HIV-1 N	9	11
SIVcpzPtt	12	15
HIV-1 P	5	4
HIV-1 O	48	48
SIVgor	6	8
SIVcpzPts	9	9
SIVmac	26	38
HIV-2	49	75
SIVsmm	32	30
SIVrcm	6	7

4. To better support the statement that amino acid preferences are different in patients than in the deep mutational scan (page 9), the authors should calculate an entropy score from the DMS data and plot it against the patient entropy scores.

In order to compute an entropy value from the DMS data we followed the methodology presented in Lee et al., 2018 in which amino acid preference at each site is calculated and Shannon entropy of the preference metric is then calculated (Lee *et al.*, 2018). We then re-scaled these entropies as Z-scores to allow fairer interpretation across the datasets. A high Z-score in the DMS dataset indicates a mutable position. We provide that analysis in Author response image 2.

Notably D773 is highly conserved in both patient dataset and has a low entropy (high preference) in the Env DMS dataset consistent with our observations of this site’s importance for Env function. Sites Y768 and L771 have higher patient conservation but have less of a preference in the Env DMS dataset. In Rev, L81, P76 and L83 all display strong preference in the Rev DMS dataset indicating that Rev is driving conservation.